# Control of the Sit-To-Stand Transfer of a Biped Robotic Device for Postural Rehabilitation

**Giuseppe Menga [1,*,†] and Marco Ghirardi [2,†]** 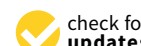

[1]  Department of Control and Computer Engineering, Politecnico di Torino, Corso Duca degli Abruzzi 24, 10129 Torino, Italy
[2]  Department of Management and Production Engineering, Politecnico di Torino, Corso Duca degli Abruzzi 24, 10129 Torino, Italy; marco.ghirardi@polito.it
[*]  Correspondence: giuseppe.menga@formerfaculty.polito.it; Tel.: +39-011-0907261
[†]  These authors contributed equally to this work.

**Abstract:** This paper deals with the control of the sit-to-stand transfer of a biped robotic device (either an autonomous biped robot or a haptic assistive exoskeleton for postural rehabilitation). The control has been synthesized, instead of considering the physiology, analyzing the basic laws of dynamics. The transfer of a human from sitting on a chair to an erect posture is an interesting case study, because it treats biped balance in a two-phase dynamic setting, with an external force disturbance (the chair–pelvis contact) affecting the center of pressure under the feet. At the beginning, a body is sitting, with a fixed pelvis moving with the hips going toward the supporting feet and, contemporaneously, releasing the load from the chair with ankles and knee torques. Then, after lift-off, it reaches and maintains an erect posture. The paper objectives are threefold: identifying the major dynamical determinants of the exercise; sythesizing an automatic control for an autonomous device; proposing an innovative approach for the rehabilitation process with an exoskeleton. For this last objective, the paper extends the idea of the authors of a haptic exoskeleton for rehabilitation. It is driven to control the joints by electromiographical signals from the patient. The two spaces, cartesian (world) and joint, where, respectively, the automatic control and the patient operate, are considered and a technique to blend the two actions is proposed. The exoskeleton is programed to perform the exercise autonomously. Then, during the evolution of the phases of rehabilitation, we postulated to seamlessly move the control from one space (purely autonomous) to another (completely driven by the patient), choosing and keeping the postural tasks and joints (heaps, knees, or ankles) on which to apply each one of the two actions without interaction.

**Keywords:** exoskeleton; haptics; rehabilitation; postural control; postural balance; multi-chain dynamical systems

## 1. Introduction

Robotic rehabilitation started with passive exoskeletons [1], or devices that impose (on the interested limbs) a forced motion. An early example for lower limbs is Lokomat [2]. It evolved into systems that create force tunnels to address the motion generated by the patient [3]. These were further extended with the introduction of a feedback from the patient, to offer cooperative controls (also called hybrid control) not only to guide but also to contribute to the efforts of the patient [4]. An extensive state-of-the-art of cooperative exoskeletons for rehabilitation is contained in [5]. Following this line of approach, we proposed a haptic exoskeleton where the joints are actuated using admittance control based on the patient's Electromiographical (EMG) signals [6].

One classical exercise for postural rehabilitation performed in a fixed position is the "sit-to-stand". Then, a haptic exoskeleton able to guide the patient to perform this exercise is highly desirable. The study of the motion of the body during this apparently simple, but in reality not so simple, exercise has attracted interest for a long time [7], not only to understand the human physiological behavior, but also to mimic the control for autonomous biped robots or for actively cooperating exoskeletons.

The majority of available studies are related to the analysis of the human physiological behavior [8,9], but also examples of synthesis of the control based on optimization are available [10,11]. A recent comprehensive review can be found in [12]. However, none discussed the key determinants at the root of the exercise. Here, we follow a different approach. Recognizing that the human motion in performing the exercise is the direct consequence of the respect of physical laws of dynamics, these laws are analyzed and a feedback control based on them is synthesized. This also offers an explanation of well know physiological results such as the "Alexander STS technique" [11].

The exercise is composed of two dynamical phases: phase 1, when still sitting on the chair, the trunk, through the hips, is moved forward to gain balance on the feet, and phase 2, when the balance is maintained moving from the chair to an erect posture. In both phases, postural balance plays a key role, however, in phase 1, the coordination between the motion of the trunk and the torques on ankles and knees to release the load from the chair are also important. From the understanding of these dynamics, an automatic control can be synthesized. However, in the case of a lower limb exoskeleton for rehabilitation, such as in [6], where an automatic postural feedback operates in the *Cartesian* space and the patient controls the joints in the *joint* space, the interaction between the two players has to be considered. This paper proposes to program the exoskeleton to perform the exercise autonomously, then, with an innovative approach, to blend the two actions, moving seamlessy during the evolution of the rehabilitation, under the direction of the physiotherapist, from purely automatic to completely under the control of the patient. Moreover, according to the needs of the rehabilitation, some of the components of the coordinates of the *Cartesian* space, indicated here as elemental postural tasks, can be actuated by the automatic control and the remaining components, through selected joints by the patient, keeping the two groups separat from postural tasks without interfering between each other. Section 2 contains a background on the dynamics of the exercise. Section 3 presents the problem, describes the general adopted model and the autonomous control. Details of control during both phases are in Section 3.3. Section 4 applies the approach to a haptic exoskeleton. The results of a simulation are discussed in Section 5. The conclusions, mentioning future ongoing researches, in Section 6 complete the paper. Details of the control algorithms are contained in Appendices A–C.

## 2. Background on the Dynamical Behaviour of the Exercise

Two main aspects characterize the exercise: balance and coordination, briefly introduced in the following subsections.

### 2.1. Balance

The dynamic of a linearized inverted pendulum is a fundamental element to understand the balance of biped systems. It was introduced by Vukobratovic [13] for controlling exoskeletons, but actually has been widely exploited in autonomous biped robotics. He argued that balance is guaranteed if the center of pressure (*CoP*) of the reaction forces exerted by the ground on the feet and is maintained in the convex hull containing the surface of the feet. This point is coincident, on a flat horizontal surface, to a point he called *ZMP* (zero moment point-where the reaction from the constraint is a pure force with zero moment), a result from classical equivalence and replacement [14] in mechanics. Moreover, a linearized inverted pendulum is a really good approximation of the more complex kinematic chain of a biped, but also, adopting this approximation, and *ZMP* motion is linked by a linear relationship to position and acceleration. In a simplified 2D environment, (the paper considers motion only on the sagittal plane) the relationship is

$$ZMP_x = COG_x - COG_z/g \cdot \ddot{C}OG_x \qquad (1)$$

where $ZMP_x$ and $COG_x$ are the motion coordinates of $ZMP$ and $COG$ on the ground, $COG_z$ is the height of the barycenter and $g$ the gravity acceleration.

Hence, to perform a transition, the $COG$ must be moved controlling the joint angles to which it is algebraically linked, having as objective the $ZMP$ position. As this requires a certain degree of anticipation, usually it is achieved by tracking some special pre-computed references of the $COG$ (preview control [15]). Choi in [16] showed that a closed loop feedback from the measures of $COG$ and $ZMP$ position is able to track a preview reference signal. This feedback has been improved in [17] by introducing in the loop a state estimator of $COG - ZMP$, extended to external disturbance forces.

*2.2. Coordination*

The biped in the sit-to-stand exercise is a dynamical system with changing non-holonomic constraints: the contacts of ground–feet are always present and the chair–pelvis is only present in the first phase only. This has consequences on the number of degrees of freedom of the multibody chain, and in the role of torque on the joints, that are defined in [18,19] as *Position* and *Auxiliary*. The former contributes to the motion, and the latter only to the reaction forces on the constraints. The balance with tracking of a preview reference of the $COG - ZMP$, based on the measures of the center of pressure under the feet and the pelvis–chair contact, and the coordination of *Position* and *Auxiliary* torques for motion and constraint force control are the elements of a "sit-to-stand" transition.

## 3. The Process, the Model and the Control

*3.1. The Process*

The exoskeleton is joined to the patient, moving on the sagittal plane with joint motion of the pairs of ankles, knees and hips of the two legs; it can be represented as a four-link chain: feet, legs, thights and trunk. The trunk, comprising head and arms, is also often called *hat*. The hips are coincident with the pelvis. In a sitting position, as is represented in Figure 1, the reaction forces transferred from the chair are applied for simplicity at the pelvis point, without exchange of torque.

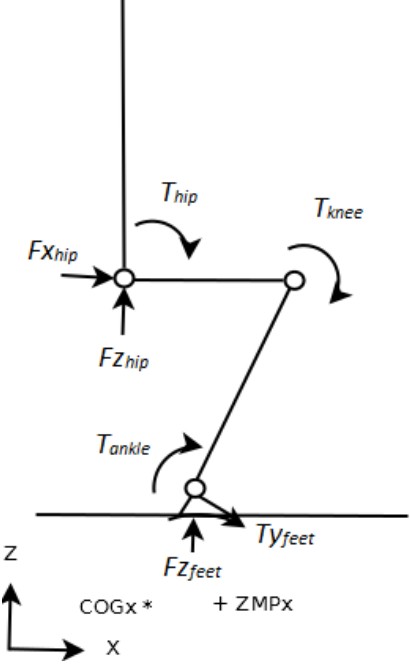

**Figure 1.** A biped sitting on a chair.

In a stationary posture, with relaxed muscles controlling ankles and knees, and the trunk attitude maintained by controlling the hips, $COG_x$ and $ZMP_x$ do not coincide as stated by the classical relationship (1), due to the presence of the external disturbance (with respect to the feet) represented by the chair reaction forces. $COG_x$ position is well behind the soles of the feet, close to the chair support, while the $ZMP_x$ alone is under the feet. The "sit-to-stand" transition starts by moving the $COG_x$ from its actual position from the chair toward the feet, tracking a preview signal which will be followed, also, during the second phase of the exercise. A typical behavior of this signal, based on Equation (1), imposing a step without overshoot of the $ZMP_x$ is given in Figure 2. Note that it depends on only one design parameter, i.e., the value of $COG_z$.

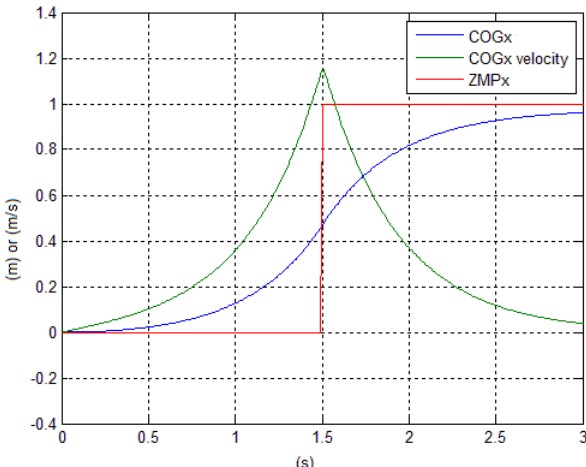

**Figure 2.** A preview signal for the $COG_x$ position and velocity imposing a step forward to the $ZMP_x$.

At the same time during phase 1, two torques are applied to the ankles and the knees to bring the reaction force under the pelvis to zero. As these torques influence the reaction forces/torques at the constraints, it is easy to see that this action, without trunk motion, would move the $ZMP_x$ back away from the feet (Figure 3), with obvious consequences on the balance.

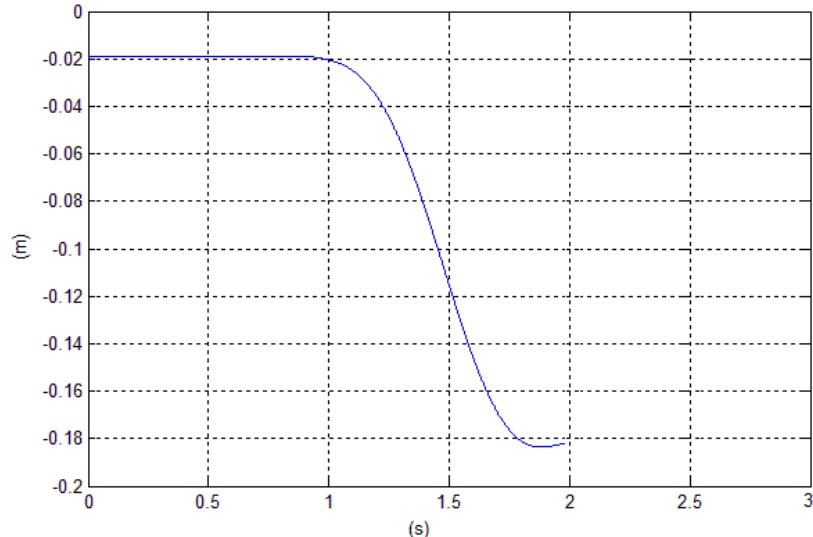

**Figure 3.** The $ZMP_x$ on the feet releasing the load from the chair, without moving forward the trunk.

The physiological behavior through proprioception, sensing the reaction forces under the feet and the pelvis, coordinates the two actions, so that during phase 1 while the reaction forces under the pelvis are zeroed, moving the trunk forward the $ZMP_x$ is approximately maintained in the original position under the feet guaranteeing balance for the next lift-off. Comparing Figures 2 and 3, (the comparison is only qualitative, as Figure 2 is generic), with transition from phase 1 to phase 2 at time 1.5 s, the behaviour of the inverted pendulum and the role of the preview can be better understood: the motion of the COG must start well in advance of any other action in order to compensate the negative effect on the $ZMP_x$ caused by releasing the load from the chair, moreover, dynamically postural equilibrium can be reached at the halfway through the transition period, even before the COG reaches the feet.

The choice of the design parameter $COG_z$ for the preview transition of the COG is not irrelevant; in fact, it can be seen in Figure 4 how the behavior of $COG_x, C\dot{O}G_x, ZMP_x$ changes if the design parameter is chosen below (red plot-with an overshoot) or above (blue plot-smoother response) the actual value (green plot) of the inverted pendulum. At last, the choice of this design parameter influences the rotational velocity of the trunk at the time of lift-off from the chair when moving to phase 2, with consequences on the kinetic energy spent and on efficiency. We argue that it is the main parameter explaining the Alexander STS technique frequently cited in the STS literature [11].

Phase 1 ends when the reaction forces from the chair become zero. At this point, the biped is in postural dynamical balance and all three joints now contribute to the motion in phase 2 to complete the exercise and reach an erect standing posture.

Obviously, these and the following are simplified observations. In reality, physiologically, through neural plasticity, almost any person who does not know anything about preview control and inverted pendulum, learns how to control the COG and the torques on the joints during the exercise optimizing, contemporaneously, the expenditure of energy.

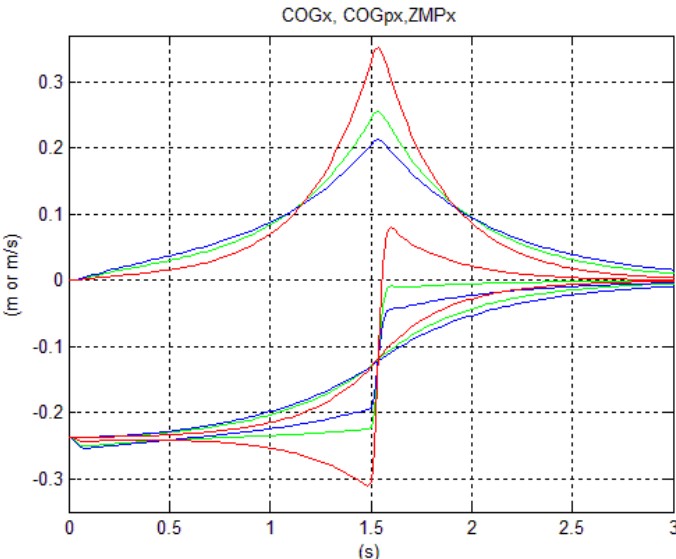

**Figure 4.** Responses of $COG_x$, $C\dot{O}G_x$ and $ZMP_x$ for previews with different parameter $COG_z$.

### 3.2. The Model

The model of the biped device in a standing up position is given in Figure 5. The kinematical chain has three degrees of freedom (DOF) with configuration variables of angles $\theta_{ankle}$, $\theta_{knee}$, $\theta_{hip}$, representing the *joint* space, with control torques $T_{ankle}$, $T_{knee}$, $T_{hip}$. The three coordinates, target variables representing three elemental postura tasks, in the *Cartesian* space, as the problem is dealing with body balance and posture, $COG_x$, the height of the pelvis and the attitude $\theta_{trunk}$ of the trunk. For simplicity, the height of the pelvis is substituted by the knee angle $\theta_{knee}$, to which, in balance, it is

strictly related. This choice also avoids singularity of the Jacobian when the knees are fully stretched. With reference to Figure 1, reaction forces from the ground to the feet, applied for convention to the vertical projection on the ground of the ankle $Ankle_x$, and torque are $F_{z_{feet}}$, $F_{x_{feet}}$ (compensated by the ground friction, is not further represented, as it has no roles) , $T_{y_{feet}}$. Forces from the chair to the pelvis (in the sitting position) are $F_{z_{hip}}$, $F_{x_{hip}}$.

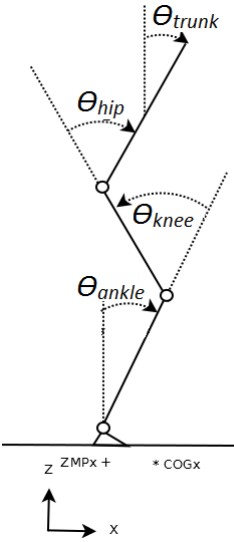

**Figure 5.** The model of the biped device.

From previous definitions, the displacement of $ZMP_x$, with respect to the position $Ankle_x$ of the conventional application of reaction forces is given by

$$\Delta ZMP_x = -T_{y_{feet}}/F_{z_{feet}}$$

i.e.,

$$ZMP_x = Ankle_x + \Delta ZMP_x \tag{2}$$

This indicates that maintaining $ZMP_x$ under the feet means keeping $T_{y_{feet}}$ close to zero.

During phase 1, in the presence of the non-holonomic constraint given by the chair–pelvis contact, the degrees of freedom of the kinematic chain are reduced: the only *Position* variable controlling the trunk attitude is the hip angle, while ankle and knee are *Auxiliary*. The latter torques control the constraint reaction forces and torques. Their mathematical expressions are composed of three terms: a nonlinear function of the angular speeds, a term generated by acceleration, and a linear function of the angular accelerations of the *Position* joints and of the control torques of the *Auxiliary* ones. Obviously, all coefficients are function of the angle values:

$$\begin{bmatrix} T_{y_{foot}} \\ F_{x_{hip}} \\ F_{z_{hip}} \end{bmatrix} = NL(\theta_{hip}, \dot{\theta}_{hip}) + G(g) + \\
\begin{bmatrix} M_{11} M_{12} M_{13} \\ M_{21} M_{22} M_{23} \\ M_{31} M_{32} M_{33} \end{bmatrix} \cdot \begin{bmatrix} T_{ankle} \\ T_{knee} \\ \ddot{\theta}_{hip} \end{bmatrix} \tag{3}$$

In the right hand side of expression (3), only the linear relationships of the control torques are of interest, which coefficients $M_{ij}$ are basically Jacobians. Target velocities in the *Cartesian* space are linked to the joint angle velocities by the Jacobian matrix $J$:

$$\begin{bmatrix} C\dot{O}G_x \\ \dot{\theta}_{knee} \\ \dot{\theta}_{trunk} \end{bmatrix} = J \cdot \begin{bmatrix} \dot{\theta}_{ankle} \\ \dot{\theta}_{knee} \\ \dot{\theta}_{hip} \end{bmatrix} \tag{4}$$

with $J$ given by

$$J = \begin{bmatrix} J_{11} & J_{12} & J_{13} \\ 0 & 1 & 0 \\ 1 & 1 & 1 \end{bmatrix}. \tag{5}$$

The Jacobians of $COG_x$, present in the Jacobian matrix with embedded knee and trunk motion (5) and the coefficients $M_{ij}$ in (3) can be generated in real time from the joint angles using expressions, which in our case, were obtained by the symbolic computational environment AutoLev [20] implementing Kane's method [14], when the kinematics is known, or better, for the $COG_x$, they can be derived from a SESC model [21] identified by a priori experiments. For details see [17].

*3.3. The Control*

Depending on the transition phase, the joint motor drivers are configured to be controlled either in velocity or in torque. The control is obtained by closing the loop using the measures of the center of pressure under the feet and the reaction forces under the pelvis, obtained from appropiate loading cells mounted on the chair and on the floor. Moreover, the position/velocities of the joints angles from the motor drivers, torques on the joints, and inertial datas from the trunk are available.

3.3.1. Velocity Control

When the joints are configured for velocity control, the inverse of the Jacobian matrix (4) of the previous section is used to generate the inputs of the motor drivers in the *joint* space to track the desired reference preview signals of the $COG_x$, of the trunk attitude and of the knee angle in the *Cartesian* space. These, during phase 2, are

$$\dot{\boldsymbol{\theta}}_{ref}(t) = \boldsymbol{J}^{-1} \cdot \boldsymbol{u}(t) \tag{6}$$

where

$$\dot{\boldsymbol{\theta}}_{ref}(t) = \begin{bmatrix} \dot{\theta}_{ankle_{ref}} \\ \dot{\theta}_{knee_{ref}} \\ \dot{\theta}_{hip_{ref}} \end{bmatrix} \tag{7}$$

is the vector of speed reference in the *joint* space and

$$\boldsymbol{u}(t) = \begin{bmatrix} u_{COG}(t) \\ u_{hat}(t) \\ u_{knee}(t) \end{bmatrix} \tag{8}$$

is the vector of the return difference signals of the feedback in the *Cartesian* space derived in the Appendix A. During phase 1, as only the hip is controlled in velocity, (6) simplifies to

$$\dot{\theta}_{hip_{ref}}(t) = J_{13}^{-1} \cdot u_{COG}(t). \tag{9}$$

### 3.3.2. Torque Control

For the drivers of ankles and knees configured to torque control during phase 1, Equation (3) are used to bring to zero the chair reaction forces, based on a recursive weighted least square scheme (described in details in Appendix B):

$$
\begin{aligned}
\boldsymbol{u_T}(t + \Delta) = \boldsymbol{u_T}(t) + \\
\alpha \boldsymbol{M}^+ \cdot (\boldsymbol{F}_{react_{ref}}(t) - \boldsymbol{F}_{react}(t))
\end{aligned}
\tag{10}
$$

where $\Delta$ is the sampling time, $u_T(t)$ is the vector of reference inputs to the torque control drivers of the motors

$$
\boldsymbol{u_T}(t) = \begin{bmatrix} T_{ankle_{ref}}(t) \\ T_{knee_{ref}}(t) \end{bmatrix}
\tag{11}
$$

$\alpha$ is a coefficient to guarantee convergence, $\boldsymbol{M}^+$ is the solution of the least square problem (A5), and $\boldsymbol{F}_{react_{ref}}(t)$ and $\boldsymbol{F}_{react}(t)$ are the vectors of reference to track and measured reaction force/torques :

$$
\boldsymbol{F}_{react_{ref}}(t) = \begin{bmatrix} T_{y_{foot_{ref}}} \\ F_{x_{hip_{ref}}} \\ F_{z_{hip_{ref}}} \end{bmatrix} , \boldsymbol{F}_{react}(t) = \begin{bmatrix} T_{y_{foot}} \\ F_{x_{hip}} \\ F_{z_{hip}} \end{bmatrix}
\tag{12}
$$

$F_{x_{hip_{ref}}}, F_{z_{hip_{ref}}}$ are chosen to send the reaction forces on the chair from their starting value to zero, and $T_{y_{foot_{ref}}}$ is kept equal to 0. The measure of $T_{y_{foot}}$ is not avilable, however it can be inferred by the distance of $ZMP_x$ from $Ankle_x$.

The choice of the weighted least square allows to blend torque transition with the need to control the $ZMP_x$. In fact, with $T_{y_{foot_{ref}}}$ close to zero $ZMP_x$ becomes fairly insensitive to the choice of the other references signals.

### 3.3.3. Lift off Transition

The control switches from phase 1 to phase 2 when the module of the reaction forces reaches a neighbourhood of zero. At that moment, a smooth transition must be imposed on the control of knees and ankles to avoid torque spikes. The simplest approach, adopted here, is to maintain (for a fraction of time at the joints) the value of torques that is reached at the end of phase 1 while the velocity control of phase 2 goes into action with a loop gain increasing from zero to the design value.

## 4. A Haptic Exoskeleton

The control described in the previous sections refers to an autonomous behavior, and it can be used to program the exercise into a biped robot, or applied in an exoskeleton in the first phase of the reabilitation when a patient is completely unable to operate.

In the case of a haptic exoskeleton partially or totally controlled by the patient, two different aspects have to be considered: the joints controlled in torque and the jonts controlled in velocity. For both aspects, EMG signals that are measured on the appropriate muscles offer approximate information of the torques applied by the patient to his joints. For the torque control, the contribution to the patient's effort is achieved with a classical technique as described in [22].

For the motion, admittance control is adopted as described in [6]. The EMG signals, processed by an admittance filter, are translated into motion information that is used as reference velocity of the corresponding joint speed drivers of the exoskeleton.

Torque control applies to phase 1 of the exercise, the contribution of the exoskeleton is to help the patient to coordinate the two actions; *motion* with the rotation of the trunk and contemporaneously *torque* for the release of the load from the chair. This can be achieved by plotting on a display in front

of the patient the position of the center of pressure in relation to the feet, with total or partial automatic support of the exoskeleton. Two options are available: automatic motion tracking a preview (different previews can be tested), and torques supported by the patient, or vice versa.

After transition to phase 2, only motion is involved, with three postural tasks, described in (4), to execute, and three joints to control. Let indicate the motion references driven by the patient's efforts, as output of the admittance filters, for ankle, knee and hip as

$$\dot{\boldsymbol{\theta}}_p(t) = \begin{bmatrix} \dot{\theta}_{ankle_p}(t) \\ \dot{\theta}_{knee_p}(t) \\ \dot{\theta}_{hip_p}(t) \end{bmatrix}. \tag{13}$$

In a training program, it is desirable that the patient progressively takes control, in number and strength, of the joints of the exoskeleton, and in so doing assumes responsibility of one or more of the postural tasks, without affecting the remaining tasks performed autonomously by the feedback. For this, the concept of *tutoring-cooperation-coordination* is introduced, where some of the elemental postural tasks, under the control of the patient, and the complementary tasks, under the automatic postural loop, can be completely decupled. This is a variant of whole body coordination (WBC) discussed in [16] and introduced in [23]. *Tutoring* means that all tasks are performed fully automatically, without intervention of the patient that is completely tutored, it is equivalent to the WBC. *Coordination* is when the patient takes complete control, through some joints, of one or more elemental postural tasks of the exercise, without interfering with the complementary tasks controlled by the automatic postural loop and, hence, he must perform coordinate motions. In between of the two extreme situations with *cooperation*, the two players can operate jointly on some tasks and their actions are overlapped. Through a modulation parameter $\beta$ the patient takes partial control and must cooperate with the automatic tasks, as depicted in Figure 6. The approach is based on two aspects: the separation of the elemental postural tasks operated by the two players, and for the tasks operated jointly, on the modulation of the two actions.

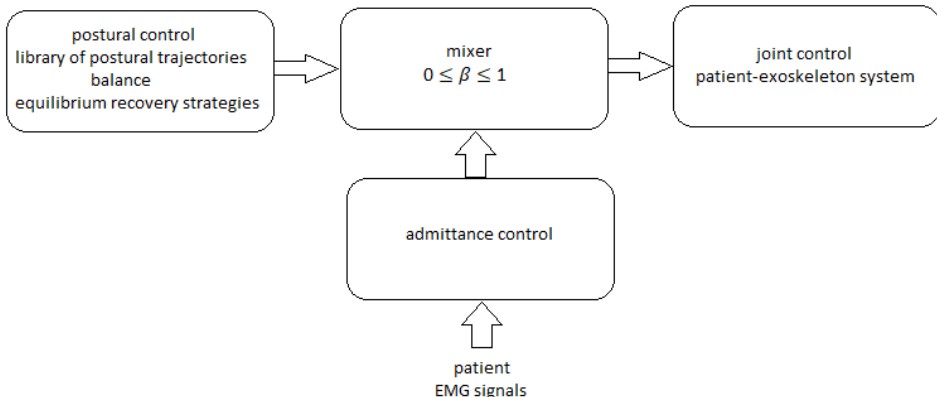

**Figure 6.** Mixing cartesian postural and patient joint controls.

Let $R_u$ represents the $3 \times 3$ selection matrix, with ones and zeros on the diagonal covering the range of the elemental tasks performed exclusively by the autonomous feedback and $N_u$ its nullspace with tasks controlled partially or totally by the patient through the joints in the range of $R_p$. As an example, when autonomous processes control $COG_x$ and trunk attitude and the patient controls the height of the pelvis through the knee, the matrices $R_u$ and $R_p$ are:

$$R_u = \begin{bmatrix} 1 & 0 & 0 \\ 0 & 0 & 0 \\ 0 & 0 & 1 \end{bmatrix}, R_p = \begin{bmatrix} 0 & 0 & 0 \\ 0 & 1 & 0 \\ 0 & 0 & 0 \end{bmatrix}, \tag{14}$$

or, the patient, through knees and hips, controls height and attitude of the trunk, with the balance, through $COG_x$, guarateed automatically, the matrices $R_u$ and $R_p$ are:

$$R_u = \begin{bmatrix} 1 & 0 & 0 \\ 0 & 0 & 0 \\ 0 & 0 & 0 \end{bmatrix}, R_p = \begin{bmatrix} 0 & 0 & 0 \\ 0 & 1 & 0 \\ 0 & 0 & 1 \end{bmatrix}. \tag{15}$$

The postural tasks in the range of $R_u$ are completely controlled by the postural feedback, while the complementary tasks in its null-space $N_u$ are jointly controlled by the feedback and by the patient, through the joints in the range of $R_p$, according to the value of a coefficient $0 \leqslant \beta \leqslant 1$. $\beta$ indicates the level of the patient's involvement in the control: *tutoring* is when $\beta = 0$, *coordination* is when $\beta = 1$, *cooperation* is when $\beta > 0 \ and \ \beta < 1$. As the coefficient $\beta$ modifies the admittance filter gain, it has, also, influence on the compliance felt by the patient on the joints in the range of $R_p$ (in fact, the joints in its null-space, not being controlled by the patient, don't offer any compliance): $\beta = 0$ completely stiff joints, $\beta = 1$ fully compliant.

The general expression for the references of the speed control of the actuators, merging automatic postural control and patient action through EMG signals, according to *tutoring-cooperation-coordination* is:

$$\begin{aligned} \dot{\boldsymbol{\theta}}_{ref}(t) &= \boldsymbol{J}^{-1} \cdot \boldsymbol{u}_p(t) + R_p \cdot \dot{\boldsymbol{\theta}}_p(t) \cdot \beta \\ \boldsymbol{u}_p &= (R_u + N_u \cdot (1 - \beta)) \cdot \boldsymbol{u} - F \cdot R_p \cdot \dot{\boldsymbol{\theta}}_p \cdot \beta \end{aligned} \tag{16}$$

where $F = R_u \cdot \boldsymbol{J}$, and with the condition

$$range(N_u) \subset range(\boldsymbol{J} \cdot R_p). \tag{17}$$

The proof of Equation (16) is presented in Appendix C.

During phase 1, while the torque control operates on ankle and knee, the *COG* tracking (9) is modified as follows

$$\dot{\theta}_{hip_{ref}}(t) = J_{13}^{-1} \cdot u_{COG}(t) \cdot (1 - \beta) + \dot{\theta}_{hip_p}(t) \cdot \beta. \tag{18}$$

Moving $\beta$ from 0 to 1, the physiotherapist increases the admittance of the joints, so that the patient gains more and more control of them. Changing $R_u$ to the zero matrix, and $R_p$ to the identity, with $\beta = 1$, the patient assumes complete control of the exoskeleton. In this condition, the autonomous control can continue to monitor the postural balance, and eventually, it inhibits incorrect patient postures, by automatically returning $\beta$ to 0.

## 5. A Simulation Example and Comparisons

In this section the simulation of the automatic control is offered. The test of the complete exercise on the exoskeleton of Figure 7 has not been performed yet. However, in [6] several examples (phase 2 only) of the transition with the cooperation of the patient using this control have been reported.

The simulation uses the same data of a patient of 75 kg, 1.8 m tall wearing an exoskeleton of 35 kg presented in [6]. The model considers only one leg and the weight of the trunk (hat) is divided by 2. So that the magnitudes of forces (N) and torques (Nm) are referred to each leg. The exercise lasts 3 s. Approximately at 1.5 s the transition from phase 1 to phase 2 occurs, with the lift off. The *COM* preview for a step transiton of the *ZMP* of Figure 2 has been chosen, testing different values of the parameter $COG_z$. Different experiments, with different degrees of difficulty, obtained with the feet in different positions with respect to the initial value of $COG_x$, i.e., to the chair, have been performed. Each position is characterized by the angle of the ankles from $10°$ to $45°$ when sitting, (i.e., feet far away from the chair and exercise more difficult, in the first case, or feet under the chair and easier exercise, in the second case). The lift-off time and transition from phase 1 to phase 2 is triggered when the module of the reaction forces on the pelvis enter into a neighbour of few Newtons of the zero. During

phase 2, simple transition functions drive knees and trunk attitude angle positions and velocities from their initial values, left from phase 1, to zero in 1.5 s. The next figures represent the most difficult experiment (experiment 1) that can be performed using this control with ankle angle of 10°. With a lower angle, e.g., 5°, the *CoP* cannot be guaranteed to remain inside the foot print. The behavior of reaction forces and torques are shown in Figure 8.

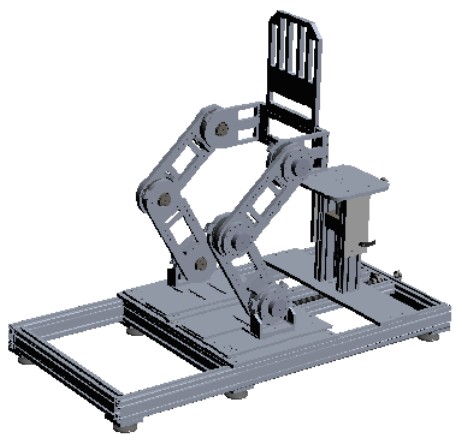

**Figure 7.** The prototype exoskeleton for the sit-to-stand exercise.

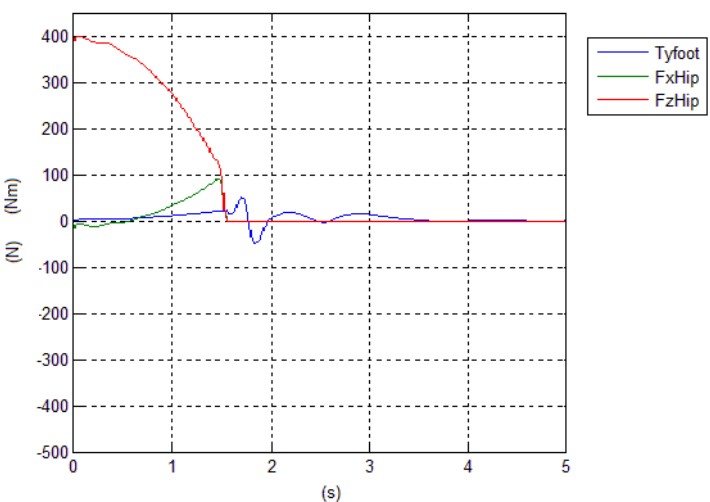

**Figure 8.** The behavior of reaction force on the hip and torque on the foot: experiment 1.

The behavior of $COG_x$ and $ZMP_x$ during the transition is given in Figure 9. The position of $Ankle_x$ is 0.

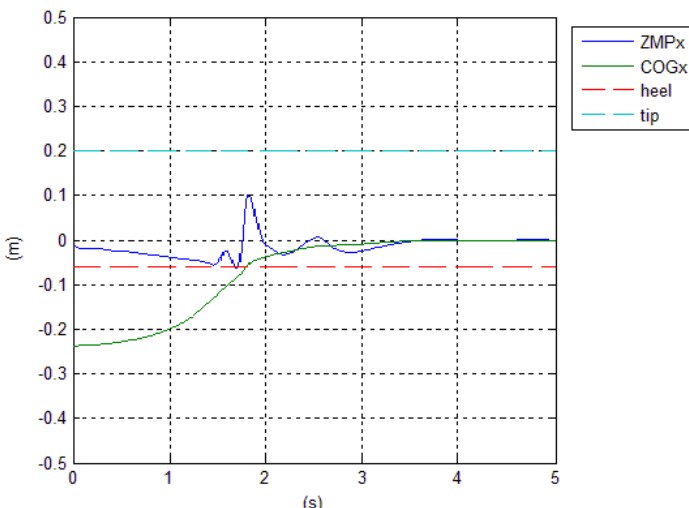

**Figure 9.** The behavior of $COG_x$ and $ZMP_x$ during the transition: experiment 1.

The torques at the joints are in Figure 10.

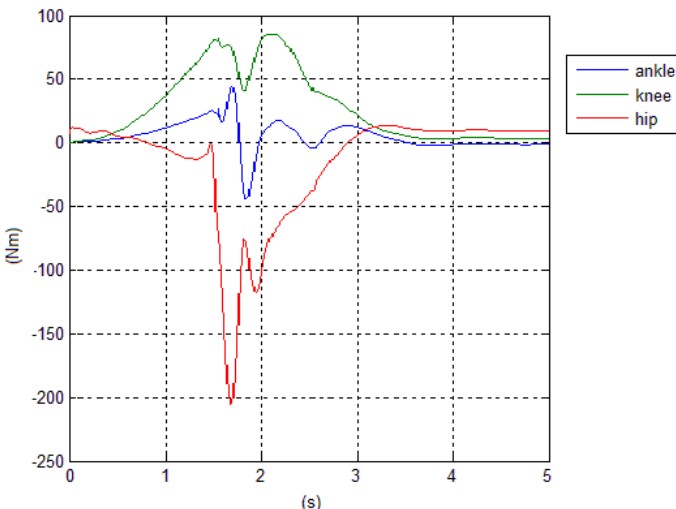

**Figure 10.** The torques at the joints: experiment 1.

Figure 11 shows the animation of the exercise.

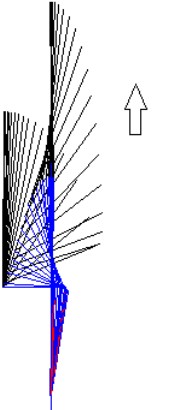

**Figure 11.** The animation of the exercise: experiment 1.

Figure 12 shows how the hat and knee angles track the references.

It is interesting to see the estimates $\hat{COG}_x$, $\hat{ZMP}_x$ theoretical assuming no disturbances and $\hat{ZMP}_x$ actual (i.e., the $CoP$), along with the estimate of the disturbances, i.e., the reaction forces on the chair, obtained by the extended estimator [17]. These are given in Figure 13.

The performance of the exercise, expressed as the ratio between the change in potential energy and the total work consumed to complete the motion, as defined in [11] is 60%, and the maximum angle reached by the trunk is 79°. As oppose, the remaining figures, from Figures 14–18, show a relatively easy exercise (experiment 2) with an ankle angle of 45°. The performance here is 95% with a maximum trunk angle value of 11°. To show the robustness of the control this exercise was performed with the identical control parameters of the previous experiment.

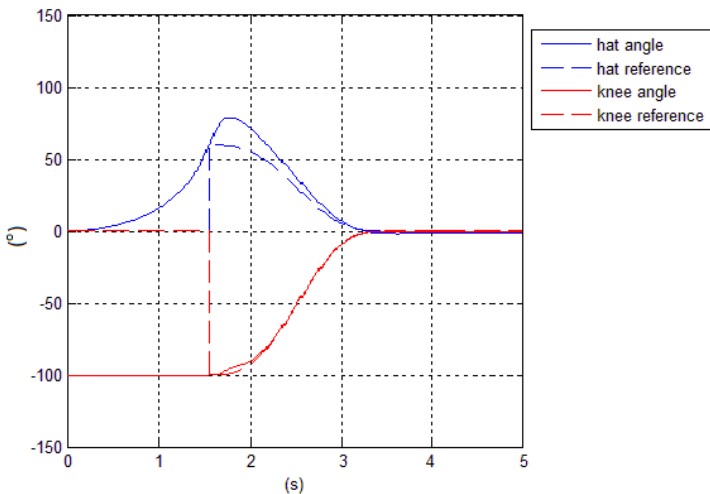

**Figure 12.** The tracking of hat and knee angles: experiment 1.

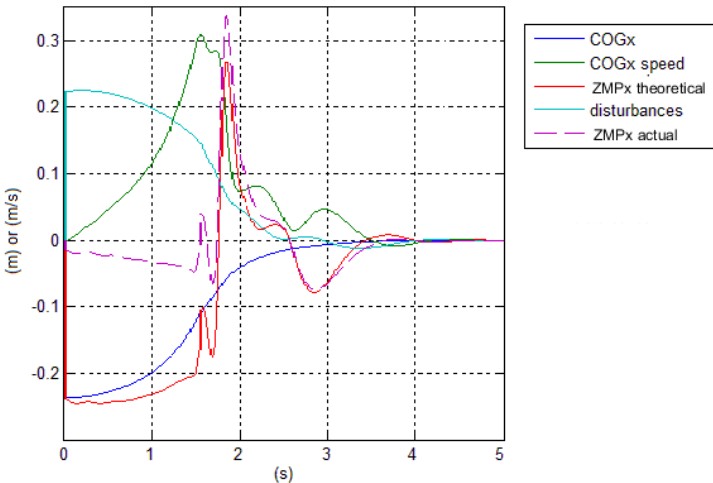

**Figure 13.** Estimates of COG and ZMP resulting from the extended estimator accounting for the pelvis-chair disturbances: experiment 1.

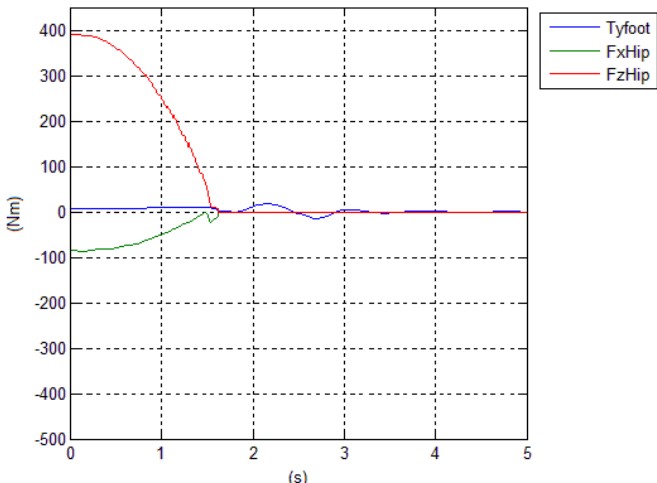

**Figure 14.** The behavior of reaction force on the hip and torque on the foot: experiment 2.

The behavior of $COG_x$ and $ZMP_x$ during the transition is given in Figure 15.

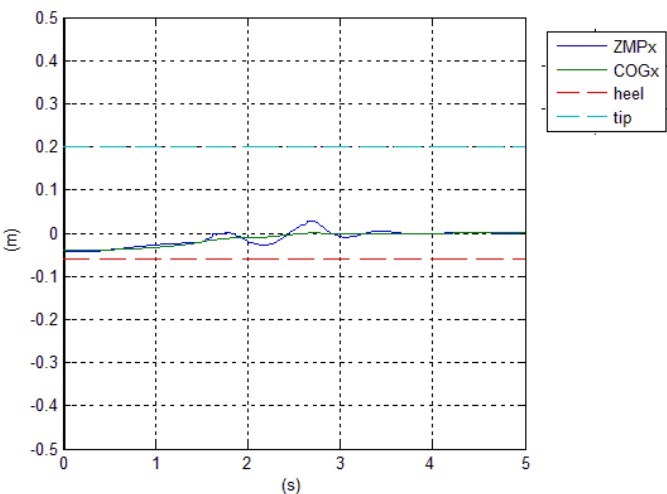

**Figure 15.** The behavior of $COG_x$ and $ZMP_x$ during the transition: experiment 2.

The torques at the joints are in Figure 16.

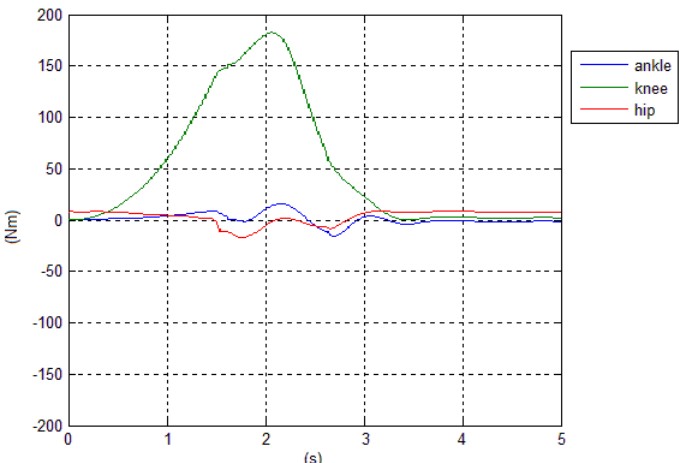

**Figure 16.** The Torques at the joints: experiment 2.

Figure 17 shows the animation of the exercise, while Figure 18 shows how the hat and knee angles track the references.

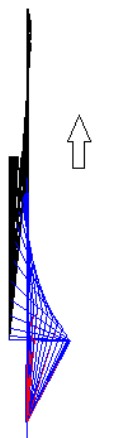

**Figure 17.** The animation of the exercise: experiment 2.

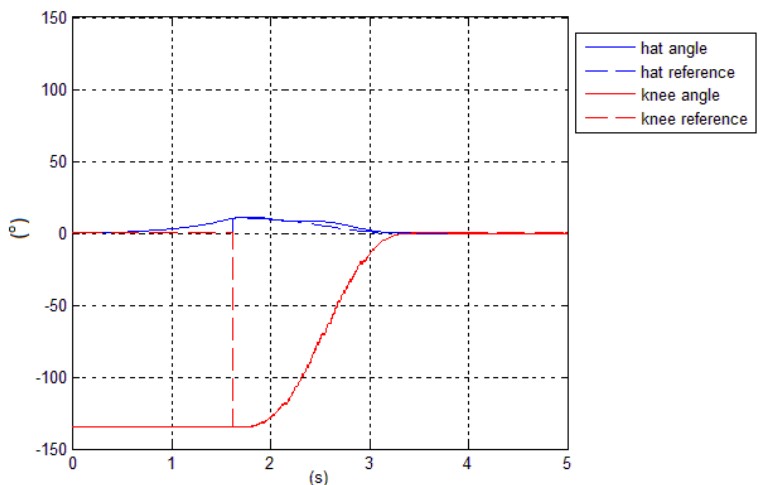

**Figure 18.** The tracking of hat and knee angles: experiment 2.

## 6. Conclusions

The paper approaches the sit-to-stand exercise under two aspects: analysis of the dynamics of the exercise/design of the feedback control of an autonomous biped device, and its application to an haptic exoskeleton with the cooperation of a patient.

In regard to the autonomous execution of the exercise, several authors have argued that two components are involved in solving the control problem in the humanoid STS motion: (1) phase and trajectory planning, and (2) feedback control.

With respect to the former, the most significant are the COM preview to follow in the transition, and the time coordination between motion of the hips and torques on ankles and knees in phase 1. From the experiments, the chair reaction forces, the knee angle and trunk attitude to reach an erect posture are not critical. The COM preview can be synthesized by exploiting the linearized inverted pendulum model, with basis the step transition of the $ZMP$, and selecting the appropiate parameter $COG_z$. The value of this parameter is not necessarily identical to the real height of the COG of the biped, but it is chosen according to the difficulty of the exercise. This difficulty here is represented by the distance of $COG_x$, i.e., the chair, from the feet when sitting. The design parameter $COG_z$ can explain the Alexander STS technique. If its value is lower or equal than the height of the $COG$, the transition is

steep, the lift off is early, the biped will be in dynamical balance, with the $COG_x$ still far from the feet, and with a relatively high trunk angular velocity to be conveyed in the successive lift motion, on the contrary if the transistion is slower, the balance will be almost statical with the $COG_x$ already under the feet, and the trunk attitude almost stationary. Based on these elements, the final trajectories can easily be planned to reduce maximum torque on the joints and energy expenditure. In the simulated experiments, two examples are considered: one is difficult with a performance, expressed as the ratio between the change in potential energy and the total work consumed to complete the motion of 60%, with the trunk reaching a maximum angle of 79°; the other is relatively easy with a performance of 95% and a maximum trunk angle of 11°.

With respect to the latter, exploiting the inverse or pseudo-inverse of Jacobian matrices and a linearized inverted pendulum model of a simple linear feedback, based on feet and chair pressure sensors, and position/velocity measures of the joint angles, can be obtained either for joint velocity or torque controls. This feedback requires a limited number of model parameters and design gains without the need to perform cumbersome inverse kinematics or dynamics. Symbolic environments such as Autolev can offer efficient expressions for the necessary Jacobians that can be computed in real time.

When the postural control is applied to an exoskeleton interacting with a patient, the paper proposes an innovative approach to blend the patient's actions with the automatic execution of the exercise. Several options are offered by the approach for proposing protocols of physiotherapy tracking the evolution of the process of rehabilitation, in terms of strenght of the support offered by the exoskeleton, and specific joints and postural tasks to be controlled by the patient. The process of rehabilitation can start with a complete *tutoring* of the patient, evolving to a *cooperation* between automatic control and patient on the same postural tasks and jonts, e.g., reaching an erect posture moving the knees, or a *coordination* between the two players operating on different tasks, e.g., the patient raises the pelvis and rotates the trunk through knees and hips while an automatic balance is guaranteed with a feedback on the ankles.

An exoskeleton for this specific exercise has been built, and some preliminary tests of phase 2 have been documented in a previous paper. However, more work is needed in future research: testing a complete exercise using the exoskeleton, not only during phase 2; decoupling the interaction not only from joints to elemental postural tasks, but also vice versa; generalizing the admittance filters from single joints and related muscles to multivariable filters processing the vector of the available EMG signals to collectively control all joints. This will be obtained exploiting muscle synergies, and training artificial neural networks that directly link EMG patterns to motion.

**Author Contributions:** Methodology, G.M. and M.G.; writing—original draft, G.M.; writing—review & editing, M.G.

**Funding:** This research has been partially supported by MIUR the Italian Ministry of Instruction, University and Research and the Piedmont Region.

**Conflicts of Interest:** The authors declare no conflict of interest.

## Appendix A. Derivation of the Velocity Controls of Section 3.3.1

The COG tracking requires a preliminary disussion of the $COG - ZMP$ estimator. Evaluations of $COG_x$ and $\dot{COG}_x$ are obtained from the joint angle measures through a SESC model [21], while $ZMP_x$ is given by measuring the center of pressure under the feet ($CoP$). These outputs, using the state estimator based on an extended inverted pendulum described in the parent paper [17], generate in real time the estimates needed to close the loop: $\hat{COG}_x$, $\hat{\dot{COG}}_x$, $\hat{ZMP}_x$ (the theoretical value in the

assumption of no disturbances) and $\hat{\delta}$, the difference between $ZMP_x$ and $CoP$, needed to take into account the chair-pelvis contact. Then, the feedback return difference signals, at each sampling time, is:

$$\boldsymbol{u}(t) = \begin{bmatrix} u_{COG}(t) \\ u_{hat}(t) \\ u_{knee}(t) \end{bmatrix} \tag{A1}$$

where $u_{COG}(t)$ refers to to the $COG$ tracking

$$u_{COG}(t) = \dot{COG}_{x_{ref}}(t) + \\ kp_1 \cdot (COG_{x_{ref}}(t) - \hat{COG}_x(t) - \hat{\delta}) \\ + kv_1 \cdot (\dot{COG}_{x_{ref}}(t) - \hat{COG}_x(t)) \\ - kz \cdot (ZMP_{x_{ref}}(t) - \hat{ZMP}_x(y) - \hat{\delta}) \tag{A2}$$

$u_{hat}(t)$ refers to to the trunk attitude tracking

$$u_{hat}(t) = \dot{\theta}_{hat_{ref}}(t) + kp_2 \cdot (\theta_{hat_{ref}}(t) - \hat{\theta}_{hat}(t)) \\ + kv_2 \cdot (\dot{\theta}_{hat_{ref}}(t) - \hat{\dot{\theta}}_{hat}(t)) \tag{A3}$$

and $u_{knee}(t)$ refers to the knee angle tracking

$$u_{knee}(t) = \dot{\theta}_{knee_{ref}}(t) + kp_3 \cdot (\theta_{knee_{ref}}(t) - \hat{\theta}_{knee}(t)) \\ + kv_3 \cdot (\dot{\theta}_{knee_{ref}}(t) - \hat{\dot{\theta}}_{knee}(t)) \tag{A4}$$

Equation (A2) was originally suggested by Choi in [16] and extended in [17], while (A3) and (A4) are basically a proportional-derivative feedback with a feedforward contribution.

During both phase 1 and 2 a preview $COG_{x_{ref}}$ of the type of those in Figure 4 has been chosen, specifically the one of Figure A1, to transfer $COG_x$ from the starting position during sitting to $Ankle_x$.

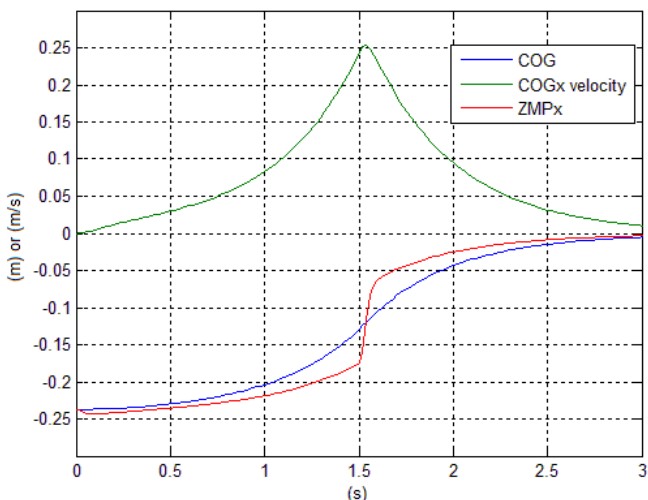

**Figure A1.** The COG preview signal for the sit-to-stand transition adopted in the examples.

For phase 2 $\theta_{hat_{ref}}$, $\theta_{knee_{ref}}$ are the references to bring the patient to a standing posture. $\theta_{knee_{ref}}$ has no special characteristics, it simply imposes a transition to the knee angle during the desired period

from sitting to standing, vice versa, $\theta_{hat_{ref}}$ is chosen to take trunk attitude position and velocity from lift-off values at the end of phase 1 to zero (Figure 12).

$k_{p_i}, k_{v_i}, i = 1 - 3, k_z$ are the feedback gains.

## Appendix B. Derivation of Torque Controls of Section 3.3.2

For torque control during phase 1, Equations (3) are used to solve the following weighted least square scheme:

$$\Lambda \cdot \begin{bmatrix} T_{y_{foot_{ref}}} - T_{y_{foot}} \\ F_{x_{hip_{ref}}} - F_{x_{hip}} \\ F_{z_{hip_{ref}}} - F_{z_{hip}} \end{bmatrix} \approx \Lambda \cdot M \cdot \begin{bmatrix} T_{ankle} \\ T_{knee} \end{bmatrix} \tag{A5}$$

and

$$M = \begin{bmatrix} M_{11} \, M_{12} \\ M_{21} \, M_{22} \\ M_{31} \, M_{32} \end{bmatrix} \tag{A6}$$

where $\Lambda$ is a diagonal matrix of weighting factors, with solution

$$\begin{bmatrix} T^*_{ankle} \\ T^*_{knee} \end{bmatrix} = \boldsymbol{M}^+ \cdot \begin{bmatrix} T_{y_{foot_{ref}}} - T_{y_{foot}} \\ F_{x_{hip_{ref}}} - F_{x_{hip}} \\ F_{z_{hip_{ref}}} - F_{z_{hip}} \end{bmatrix} \tag{A7}$$

$$\boldsymbol{M}^+ = (M^T \Lambda^2 M)^{-1} \Lambda^2 M^T, \tag{A8}$$

## Appendix C. Proof of Equation (16)

Consider the building block diagram Figure A2. It implements Equation (16), and shows the generation of the $\dot{\boldsymbol{\theta}}_{ref}$ with both autonomous control and patient's contribution. With block diagram manipulations advance the patient's contribution $\dot{\boldsymbol{\theta}}_p$ before $\boldsymbol{J}^{-1}$ from *joint* to *Cartesian* space.

The speed signals in the Cartesian space result as

$$(R_u + N_u \cdot (1 - \beta)) \cdot \boldsymbol{u} + (\boldsymbol{I} - R_u) \cdot \boldsymbol{J} \cdot R_p \cdot \dot{\boldsymbol{\theta}}_p \cdot \beta \tag{A9}$$

It should be noted that $\boldsymbol{I} - R_u = N_u$, then

$$(R_u + N_u \cdot (1 - \beta)) \cdot \boldsymbol{u} + N_u \cdot \boldsymbol{J} \cdot R_p \cdot \dot{\boldsymbol{\theta}}_p \cdot \beta \tag{A10}$$

Condition (17) assures that the action of the patient on the joints in the range of $R_p$ encompasses all postural tasks not covered by the automatic control.

Equation (A10) can be interpreted as follows:

- If $\beta = 0$, all tasks are controlled by the feedback loop;
- If $\beta = 1$, tasks in $R_u$ are controlled by the feedback loop, while tasks in $N_u$ are controlled by the patient, that, obviously must coordinate his action with the other tasks;
- If $0 < \beta < 1$, tasks in $R_u$ are controlled by the feedback loop, while tasks in $N_u$ are jointly controlled by the feedback loop and by the patient, blended by the coefficient $\beta$.

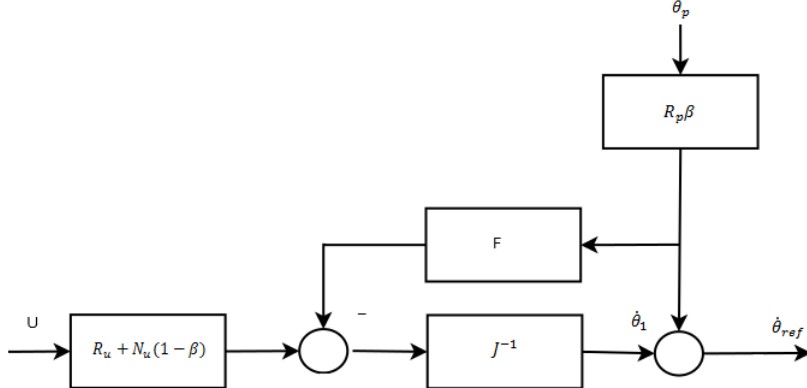

**Figure A2.** Integrating autonomuos control and patient action in controlling the joints of the exoskeleton.

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
