# Peer review of "Control of the Sit-To-Stand Transfer of a Biped Robotic Device for Postural Rehabilitation"

_robotics, doi:10.3390/robotics8040091_

Round 1
Reviewer 1 Report
It is clear that the authors devoted significant time and energy to this paper. That being said, there are a number of things that need to be corrected before this paper can be considered for possible publication.
First, the authors need to make the focus of the paper more clear. The exact nature of what they are proposing should be more accurately described. Both autonomous robots and human assistive exoskeletons are described; however, it is unclear that these would have the same exact operations or that assumptions that are applied would apply to both.
Second, the authors should illustrate the specific concepts of what they are describing using models that depict a robot and a human with an exoskeleton. The line diagram can also be presented and should, perhaps, be superimposed over the photographs or renders of the assistive technology / human and robot images to provide context.
Third, the paper is too equation driven. Readers should be able to follow the paper without having to analyze lots of equations. Putting some of the equations in an appendix has already been done. More should be removed to make the paper an easier read for readers and the equations that remain should be explained in the text so that the reader can follow the paper without having to stop -- if they don't want to -- and try to figure out the significance and relevance of each equation. Also, it is not necessary to show every step related to using the equations, just focus on illustrating major steps and results.
Forth, what is shown in the graph figures should be better explained. The importance of this data should be explained and it should be analyzed.
Fifth, the conclusions section should be revised to focus on what the contribution of this paper was and presenting the key results of the analysis of your data. There is too much focus on explaining prior work in this area now. The conclusion should be more focused.
Sixth, work to cut down the use of acronyms. Try to focus on have a few acronyms that are very frequently used. Having too many requires readers to have to figure out what the acronym means while trying to read the paper, which can be disruptive.
Seventh, the COG acronym is not defined properly. The acronym is put into parenthesis proximal to a relevant discussion but it doesn't include what it stands for.
Overall, it is clear that a lot of work has been done. This paper needs more context about what is being proposed in order to be able to be fully assessed. Making the changes described above should facilitate further review with greater context being considered.
I would be happy to review a revision of this paper, if asked.
Reviewer 2 Report
This paper presents the problem of the control of the sit-to-stand transfer of a biped robotic device. The Authors state that none discusses the key determinants at the basis of the task and in the presented work, a different approach is proposed: recognizing that human behavior is the direct consequence of the respect of physical laws of dynamics, these laws are analyzed and a feedback control is synthesized based on them.
For the reviewer is not clear, if the Authors want to apply the control theory developed in the field of bipedal humanoids robots to the exoskeletons for rehabilitation. If this is the reading key, the Review suggest making clear sentence about the motivation of the paper in the abstract. Then, the introduction will defend the abstract providing evidence based on the state of the art. To me, abstract if too deep in the technical solution, and the introduction is confusing by taking control theory from bipedal humanoids and rehabilitation exoskeletons without giving already clear vision to the reader.
The experimental section is not clear if the proposed control have been applied in the previous publication [6], or the authors are describing based on the data gathered if this presented approach may work. Since the device is available to the authors, a specific experimental section may be conducted to compare the effectiveness of this control approach respect to a different one (based on literature).
Round 2
Reviewer 1 Report
It is very clear that the authors have devoted significant work to revising this paper. That being said, there are a number of additional revisions that need to be made before this paper can be further reviewed for possible publication.
First, there are still a number of English usage issues. Please have someone more fluent in English review the paper and make changes, as needed.
Second, the listing of topics in the abstract makes things clearer. I would suggest removing the "1)" etc. and replacing them with "first" and such.
On line 28, what is being referred to by "and the contained bibliography" is unclear.
On lines 39-40, the statement "that human behavior is the direct consequence of the respect of physical laws of dynamics" may perhaps be better stated as "human movement is the ..."
The extensive use of sub-sections in section 1 is problematic. Readers are trying to find the research statement in the introduction. With this section running onto the third page, it makes locating the thesis of the paper problematic. Also, it is unclear as to exactly where this is located in the current configuration of this section. II would suggest that this section should be split into two sections with the new section 1 focusing on introducing the paper and making the thesis or research statement. Other background materials can be located in the new section two.
In line 86, the "1)" and "2)" do not need the ")". This similar issue exists in line 106 and other locations and should be corrected in all locations.
In lines 234-235, the wording is confusing and it is unclear as to exactly what the authors mean.
Lines 276-278 are confusing and should be reworded.
In your response to previous comments, you mention that a picture can be found in ref. 6. Figures 1 and 14 should be included in this paper and cited back to the earlier paper, as they would dramatically increase understanding of what you are describing. Figure 2 or something super-imposed over one of the other figures may be useful too. As this article was published in this same journal there would be no copyright issues with using these figures. Readers should not have to track down this additional paper for the current one to be clear.
Finally, there is no section header for the references section.
This an interesting paper and clearly has been enhanced by the authors changes. These additional changes should further enhance the paper and prepare it for publication.
I would be happy to review a revision, if asked.
Reviewer 2 Report
Thank you for revising the paper; I believe the revised manuscript is
more interesting and easier to follow. However the Reviewer suggests to
underline in conclusions numerical achievements (estimation error,
controllability) and the open issue that will be addressed in future
work (such as experimental evaluation).
